# Horizontal antimicrobial resistance transfer drives epidemics of multiple *Shigella* species

Kate S. Baker [1,2], Timothy J. Dallman[3], Nigel Field[4], Tristan Childs[5], Holly Mitchell[4], Martin Day[3], François-Xavier Weill[6], Sophie Lefèvre[6], Mathieu Tourdjman[7], Gwenda Hughes[8], Claire Jenkins[3] & Nicholas Thomson[2,9]

Horizontal gene transfer has played a role in developing the global public health crisis of antimicrobial resistance (AMR). However, the dynamics of AMR transfer through bacterial populations and its direct impact on human disease is poorly elucidated. Here, we study parallel epidemic emergences of multiple *Shigella* species, a priority AMR organism, in men who have sex with men to gain insight into AMR emergence and spread. Using genomic epidemiology, we show that repeated horizontal transfer of a single AMR plasmid among *Shigella* enhanced existing and facilitated new epidemics. These epidemic patterns contrasted with slighter, slower increases in disease caused by organisms with vertically inherited (chromosomally encoded) AMR. This demonstrates that horizontal transfer of AMR directly affects epidemiological outcomes of globally important AMR pathogens and highlights the need for integration of genomic analyses into all areas of AMR research, surveillance and management.

[1] Institute for Integrative Biology, University of Liverpool, Liverpool L69 7ZB, UK. [2] Wellcome Trust Sanger Institute, Hinxton CB10 1SA, UK. [3] Gastrointestinal Bacterial Reference Unit, National Infection Service, Public Health England, London NW9 5HT, UK. [4] Centre for Molecular Epidemiology and Translational Research, Institute for Global Health, UCL, London WC1E 6BT, UK. [5] Centre for Infectious Disease Surveillance and Control, National Infection Service, Public Health England, London NW9 5HT, UK. [6] Institut Pasteur, Unité des Bactéries Pathogènes Entériques, Paris 75015, France. [7] Santé Publique France, the French Public Health Agency, Saint-Maurice 94415, France. [8] Department of HIV and STIs, National Infection Service, Public Health England, London NW9 5HT, UK. [9] London School of Hygiene and Tropical Medicine, London WC1E 7HT, UK. Correspondence and requests for materials should be addressed to K.S.B. (email: kbaker@liverpool.ac.uk) or to N.T. (email: nrt@sanger.ac.uk)

Bacterial genomics has revealed that antimicrobial resistance (AMR) phenotypes are conferred largely by either de novo mutation in the chromosome or the acquisition of new AMR genes through horizontal gene transfer (HGT), both of which can then be vertically propagated. The importance of HGT in the current global AMR crisis is evidenced by repeated observations of a new AMR gene in one organism in one location being followed by global reports of the gene in other organisms and geographical areas. This has happened for the carbapenemase-encoding $bla_{NDM-1}$ gene, the extended spectrum beta-lactamase encoding $bla_{CTX-M}$ genes, and is currently happening for mcr-1, the gene encoding resistance against the last-line antimicrobial, colistin[1–3]. Despite this inferred importance of HGT in global AMR emergence, the real-time epidemiological impact of HGT on AMR emergence has never been observed so is poorly understood. Observations have been made that are suggestive of HGT of AMR (i.e., similar or identical AMR plasmids in multiple bacterial species within a hospital or patient)[4,5], and such evidence would be enhanced by strong epidemiological data to support the plausibility of HGT occurring, and information on the epidemiological fate (e.g., seeding an outbreak, or a dead-end transmission) of the newly formed pathogen-AMR combination. Thus, to understand the impact of HGT in the emergence of new antimicrobial resistant pathogens, it is essential that we study HGT alongside associated epidemiological information, and at the population level, where epidemiological outcomes are less stochastic.

*Shigella* in MSM is a relevant model for studying AMR emergence owing to features of both the pathogen and its AMR, as well as features characteristic of the epidemiological community of MSM. Shigellae cause a large burden of disease and are globally distributed. They are one of the most common causes of childhood diarrhoeal disease in low-to-middle income nations and also cause disease in high income nations often, but not always, linked with travel[6,7]. In addition to its concerning disease burden, shigellae are increasingly resistant to antimicrobials (having accumulated horizontally acquired AMR genes over time)[8,9] and ciprofloxacin-resistant *Shigella*, a phenotype conferred by mutations in the Quinolone Resistance Determining Regions (QRDR) of the chromosome, is recognised by the World Health Organisation as a priority AMR pathogen[10]. Within the epidemiological community of MSM in England there are dense transmission networks of individuals with similar risk profiles (e.g., HIV co-infection, high partner numbers) currently permissive of pathogen epidemics, including some that are highly resistant to antimicrobials. This permissiveness is evidenced by recent, simultaneous epidemics of multiple sexually transmissible illnesses (STIs) in this community, including Lymphogranuloma venereum, syphilis, and gonorrhoea, the latter of which has become highly multi-drug resistant[11–16].

In addition to these more traditional STIs, *Shigella* has emerged as a sexually transmissible enteric infection in MSM that is capable of global dispersal and is associated with AMR[17,18]. An MSM-associated (MSMA) *S. flexneri* 3a lineage that emerged epidemically in England in 2009 has since disseminated across four continents through international transmission among MSM[19,20], and successful sublineages of this MSMA lineage carried azithromycin resistance on the conjugative plasmid pKSR100[19]. Since 2012, England has experienced two further, parallel epidemics of domestically acquired shigellosis thought to be associated with MSM (caused by another serotype of *S. flexneri;* 2a, and another species, *S. sonnei*[21]).

Here, we study these parallel *Shigella* epidemics in a densely connected epidemiological community to elucidate the role of HGT on the epidemiology of AMR emergence and spread. We show that HGT of a single azithromycin resistance plasmid

enhanced and facilitated epidemic emergence of multiple *Shigella* species. This demonstrates the importance of tracking the genetic context of AMR within pathogen populations.

## Results

**Co-circulating MSMA sublineages**. To identify MSMA sublineages, an epidemiologically representative random subsample of *S. flexneri* 2a and *S. sonnei* isolates and associated epidemiological metadata were selected from the national archive of isolates held by the national reference laboratory at Public Health England collected between 2008 and 2014. The national archive contains all *S. flexneri* and ~ 70% of *S. sonnei* isolated in hospital, private and regional laboratories (as onward reference laboratory submission is recommended for *S. sonnei*, but mandatory for *S. flexneri*), and basic epidemiological data were extracted from sample submission forms, i.e., region, gender, patient age and travel history where noted (as in[21,22]). The time-period selected encapsulated three parallel epidemic emergences of *Shigella* sp. in England, and isolates were subsampled proportionately to the number of cases by year with a greater weighting of domestically acquired infections (non-travel-associated isolates) and patient age stratification (16–60 year olds only) for *S. sonnei* owing to the large number of isolates of this species (Table 1, Supplementary Data 1, Supplementary Table 1). The isolates ($n = 366$) were whole-genome sequenced (as in ref. [17]) and whole-genome phylogenies were constructed (Fig. 1). Correlating isolate phylogenetic positions with patient characteristics revealed six sublineages in which 16–60 years old, male patients without a history of recent high-risk travel (i.e., travel to Africa, Asia or Latin America) were significantly over-represented (Odds ratio (OR) 17.8 95% confidence interval (CI) 8.0–46.6, $p = 2.2e-16$, Fig. 1, Methods, Supplementary Table 2). These demographic characteristics are typical of MSMA shigellosis patients, and these six sublineages were designated MSMA. This MSMA designation was confirmed through a significant association ($p = 0.0140$, Fisher's exact test) of isolate designation to an MSMA sublineage with recent (within 1 month of diagnosis) patient MSM activity (Methods). The MSMA sublineages were designated major and minor in *S. flexneri* 2a and 1–4 for *S. sonnei* (Fig. 1). Crucially, sublineage presence over time and temporal phylogenetic reconstruction indicated substantial periods of co-existence of sublineages (Fig. 2). In addition to being co-existent, the MSMA sublineages were also co-located in geographical space, with at least 38% of isolates in each MSMA sublineage being from patients in London (Supplementary Data 1). Thus, the six MSMA sublineages were circulating in the same epidemiological community (i.e., London MSM) as each other, alongside the intercontinentally transmitted MSMA *S. flexneri* 3a[19].

**Population-level success of MSMA sublineages**. Although MSMA sublineages were co-circulating their population-level outcomes differed. Using the combined genomic and epidemiological information, we defined several markers of population-level success. Specifically, these were: a higher relative disease burden; greater longevity/persistence of a sublineage in the MSM population over time; and evidence of international transmission.

Some MSMA sublineages had a higher disease burden in the English setting (Fig. 2) relative to other MSMA sublineages. For example, the minor sublineage of *S. flexneri* 2a contained just seven isolates, whereas the major sublineage contained 47 isolates. As these isolates are a representative subsample of cases over time, proportional differences between coexisting sublineages are indicative of different population-level success. Taking the number of isolates attributable to a given MSMA sublineage per year as a crude estimate of population-level success suggests that

### Table 1 Shigella sp. isolates used in this study

| Species/Serotype | Isolate set | Reference | Selected | Function in this study | Number |
|---|---|---|---|---|---|
| S. flexneri 2a | English representative subsample, 2008–2014 | This study | Representatively 9% (32 of 368 cases) of travel, 18% (147 of 820 cases) of non-travel-associated cases | Primary data set under study | 179 |
| | French representative subsample, 2009–2014 | This study | Representative of isolates received | Determine whether English MSMA sublineages were present internationally | 40 |
| S. sonnei | English representative subsample, 2008–2014 | This study | Representatively 3% (45 of 1539 cases) of travel, 8% (142 of 1751 cases) of non-travel-associated cases. Age-restricted (16–60 years old) to explore circulation in adults. | Primary data set under study | 187 |
| | French representative subsample, 2009–2014 | This study | Representative of isolates received | Determine whether MSMA sublineages were present internationally | 75 |
| | UK 2015/16 MSMA clusters | 22 | Identified as MSMA clusters, see reference | Determine whether MSMA sublineages identified in English representative subsample remained in circulation in 2015/16 | 50 |
| S. flexneri 3a | MSMA sublineage isolates | 19 | Representatively 20% of all isolates, see reference | Used to examine the horizontal transmission of pKSR100 | 206 |
| | | | | Total | 737 |
| | | | | Total (this study) | 481 |

English isolates were supplied by the Gastrointestinal Bacteria Reference Unit at Public Health England, London. For the English representative subsample, non-travel-associated isolates were intentionally over-represented and travel-associated isolates were included to provide temporally-relevant context. French isolates were supplied from the French National Reference Centre for E. coli, Shigella and Salmonella at the Institut Pasteur, Paris

the major sublineage of S. flexneri 2a and sublineage 2 of S. sonnei generated the greatest disease burden for each species, respectively, (cases/year in Fig. 2, NB: comparison between species is invalid owing to different selection criteria—see Table 1, Supplementary Table 1, and ref. [19]). For S. sonnei, it was also possible to explore persistence of MSMA sublineages in England over time. This persistence was evaluated by combining the 2008–2014 representative subsample with data from isolates of MSMA shigellosis clusters detected England in 2015/2016[22] (n = 50, Table 1, Supplementary Data 1). Of these 50 isolates from 2015/2016, 12 clustered with sublineage 2 and 32 clustered with sublineage 4, indicating that these two sublineages had continued to transmit in the English MSM population (Fig. 2, Supplementary Fig. 1).

In addition to those markers of population-level success in English MSM, whether these MSMA sublineages were present in a nearby nation with high travel connectivity (France) was assessed. The French and English MSM epidemiological communities are known to be connected as they were previously shown to have phylogenetically admixed isolates of the intercontinentally distributed sublineage of MSMA S. flexneri 3a[19]. The 2008–2014 representative subsample from England was compared with a similarly representative subsample of S. flexneri 2a and S. sonnei from France collected between 2009 and 2014 (n = 115, Table 1, Supplementary Data 1). These analyses showed that four French isolates belonged to the major sublineage of S. flexneri 2a, and 13 and 10 French isolates clustered with sublineages 1 and 4 of S. sonnei respectively, indicating the presence of these three sublineages in France (Fig. 2, Supplementary Fig. 1). Notably, isolates belonging to MSMA sublineages from France were also predominately (26 of 27) from 16–60 years old male patients with no recent history of high-risk travel (Supplementary Data 1).

Collectively, these markers of population-level success suggest that the minor sublineage of S. flexneri 2a and sublineage 3 of S. sonnei performed poorly in their natural transmission setting relative to the other MSMA sublineages. In addition to causing a comparatively low number of cases and not persisting through to 2015–16 in UK MSM, they were not found in France, and were also the only two MSMA sublineages not observed in the final years of the sampling window, despite having been the earliest two sublineages in existence (Fig. 2).

**Associations with AMR**. To investigate the association of population-level success with AMR, genetic determinants of AMR were characterised and correlated with phylogeny. Specifically, this included the acquired resistance gene content and the presence of QRDR mutations (identified as in ref. [23]). For both species, vertically inherited triple QRDR mutations responsible for resistance to ciprofloxacin were exclusively associated with monophyletic sublineages related to travel from Asia (OR 98 95% CI 29.7–395, p < 0.0001, Fig. 1, Supplementary Data 1), although one S. sonnei MSMA sublineage was nested within this ciprofloxacin-resistant sublineage. Unlike the MSMA sublineages, the occurrence of these ciprofloxacin-resistant sublineages over time was comparatively stable, or only slightly increasing (Fig. 2, Supplementary Data 1). In contrast, horizontally acquired resistance genes encoding azithromycin resistance were present multiphyletically, and were strongly associated with MSMA sublineages (OR 58, 95% CI 20–231, p < 0.0001, Fig. 1, Supplementary Data 1). Genes encoding ESBLs were uncommon and not associated with specific phylogenetic sublineages (Fig. 1). Other AMR determinants conferring resistance to beta-lactams, trimethoprim, sulphonamide, chloramphenicol, tetracycline, streptomycin and spectinomycin antimicrobials were common among

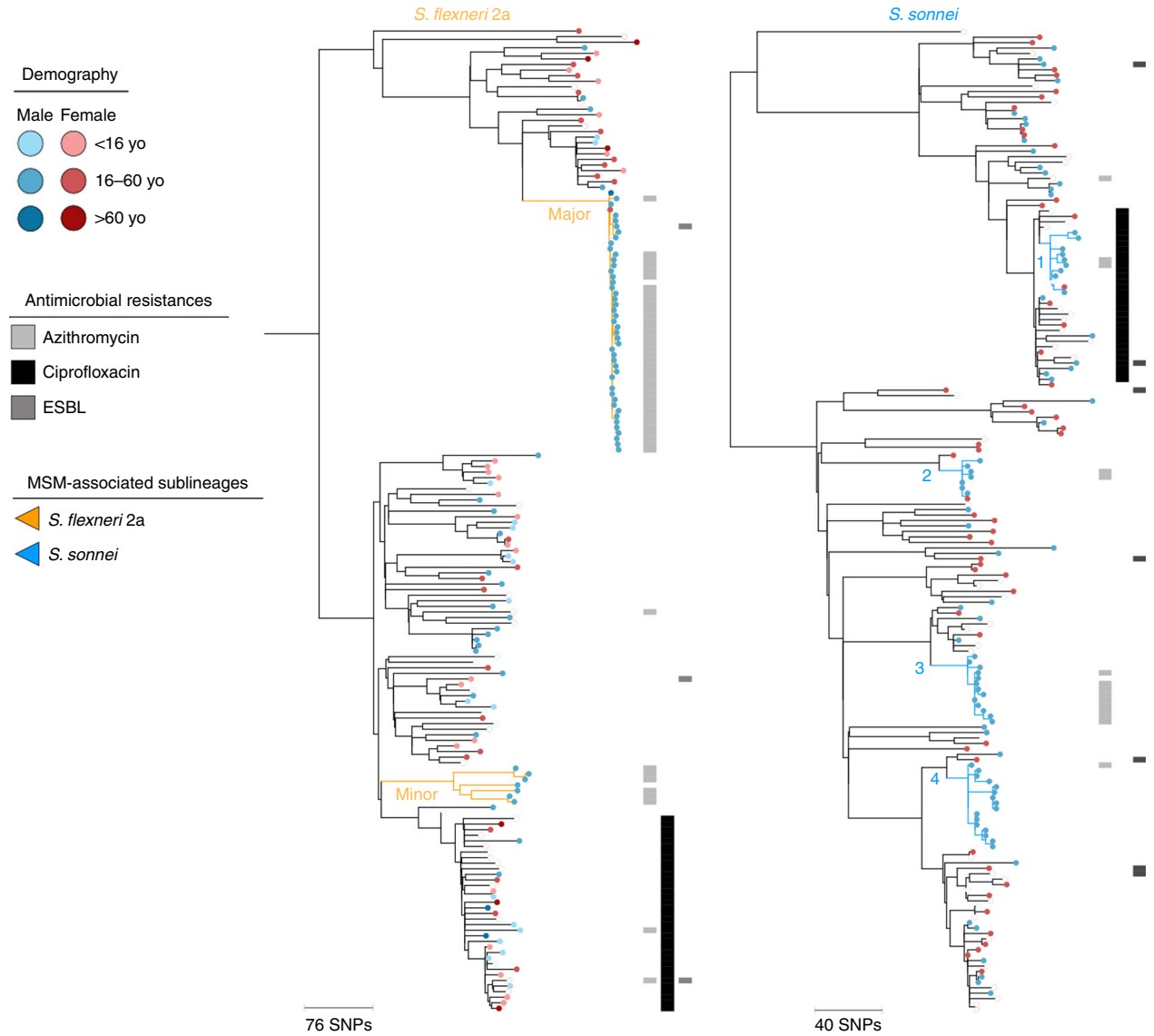

**Fig. 1** MSM-associated sublineages and antimicrobial resistance associations in context. The midpoint rooted maximum likelihood phylogenetic trees are shown for PG3 of *S. flexneri* 2a and Lineage III of *S. sonnei*. Scale bars in single nucleotide polymorphisms (SNPs) are shown below each phylogeny. Patient demography of isolates is shown overlaid on tips according to the inlaid key with circles of patients reporting recent travel to regions endemic for shigellosis being hollow. MSM-associated sublineages are labelled and coloured according to the inlaid key. Anticipated resistance phenotypes for azithromycin, ciprofloxacin (only triple QRDR mutants), and ESBL are shown in adjacent tracks

the isolates (Supplementary Data 1) and frequently carried on mobile genetic elements. Notably, the association of MSMA shigellosis with azithromycin resistance (rather than resistance to ciprofloxacin, the recommended treatment for shigellosis[24]) is thought to have arisen consequent to high treatment rates for other STIs commonly found in the shigellosis-affected MSM subpopulation[15,16,19,25], rather than from empirical treatment for shigellosis. This 'collateral damage' hypothesis is supported by recent treatment reviews (both from the United Kingdom and the United States), which suggest that upwards of 80% of MSM with shigellosis receiving antimicrobial therapy are treated with ciprofloxacin[18,26].

**Fit sublineages are associated with pKSR100.** To better understand the association of azithromycin resistance with MSMA shigellosis, we fully characterised the genetic elements carrying azithromycin-resistance genes in each MSMA sublineage (Methods, Supplementary Fig. 2). Azithromycin-resistance genes

were always carried by large (>70 kb) plasmids, and the major sublineage of *S. flexneri* 2a and sublineages 1, 2 and 4 of *S. sonnei* carried the resistance genes *mphA* and *ermB* on the same plasmid; pKSR100, which contributed to the intercontinental emergence of MSMA *S. flexneri* 3a (Fig. 2, Supplementary Fig. 2, Supplementary Fig. 3). To confirm the association of pKSR100 with MSMA sublineages regardless of clonal population structure, and refute the existence of association with other genetic determinants, a bacterial genome-wide association study to detect sequence elements significantly associated with MSMA sublineage isolates was conducted (Methods), which confirmed that pKSR100 was strongly associated with MSMA sublineages (Fig. 3). The only other contiguous sequence to contain any sequence element with a higher significance than pKSR100 had a reduced density of MSMA-associated sequence elements and a significantly lower mean associative value relative to pKSR100 (Supplementary Fig. 4). The azithromycin-resistance plasmid in the minor sublineage of *S. flexneri* 2a and sublineage 3 of *S. sonnei* were distinct

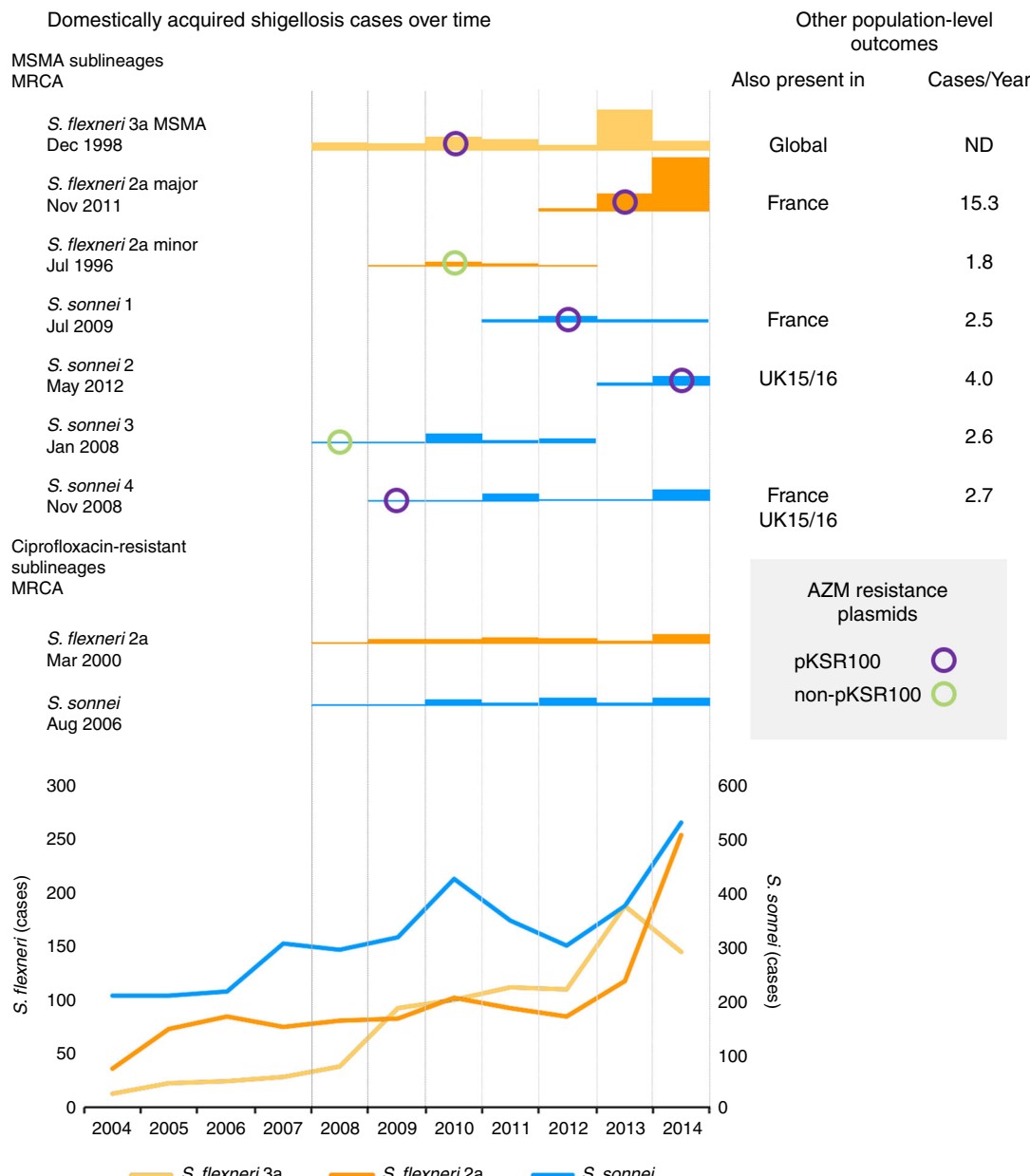

**Fig. 2** MSMA sublineage epidemiology and relationship with azithromycin (AZM) resistance determinants. The total number of domestically acquired shigellosis cases is shown in the below graph. Above, the relative number (owing to differences in sampling approach, *S. flexneri* 3a is scaled 1:3 relative to the other sublineages, exact numbers available in Supplementary Data 1) of isolates sequenced that belonged to different MSMA and ciprofloxacin-resistant sublineages is shown by year, along with the time to most recent common ancestor (MRCA). These bar charts are overlaid by a marker indicating the year of first observation, and classification, of azithromycin-resistance plasmids in the sublineage, according to the inlaid key. For MSMA sublineages, other population-level outcomes, as discussed in the text, are also shown to the right, including other regions and times where the sublineage was detected (also present in) and the number of cases detected per year (cases/year) attributable to that sublineage

from pKSR100 and carried only *mphA* for encoding azithromycin resistance (Fig. 2, Supplementary Fig. 2). Phenotypic testing on a subset of isolates revealed that these plasmids conferred differing levels of azithromycin resistance depending on resistance gene content. Specifically, isolates carrying both *mphA* and *ermB* genes ($n = 17$) had minimum inhibitory concentrations (MICs) of $\geq 64$ mg/l, whereas isolates containing only the *mphA* gene ($n = 13$) had MICs $\leq 32$ mg/l and isolates with neither gene ($n = 43$) had MICs of $\leq 2$ mg/l (Supplementary Data 1, Methods, Supplementary Table 3).

Within MSMA sublineages, pKSR100 was present in those sublineages that also had some makers of population-level success

and the plasmid also appeared to have a role in driving epidemics. For example, the major *S. flexneri* 2a sublineage and the intercontinental *S. flexneri* 3a MSMA lineage, both showed a dramatic increase in numbers following the introduction of pKSR100 into those sublineages (Fig. 2, Supplementary Data 1, and ref. [19]). Similarly, *S. sonnei* MSMA sublineages that were still present in the MSM transmission network in 2015/2016 and/or were present in France (i.e. sublineages 1, 2 and 4) also contained pKSR100. Contrastingly, the two MSMA sublineages that did not contain pKSR100 (the minor sublineage of *S. flexneri* 2a and sublineage 3 of *S. sonnei*) did not achieve such markers of population-level success (Fig. 2).

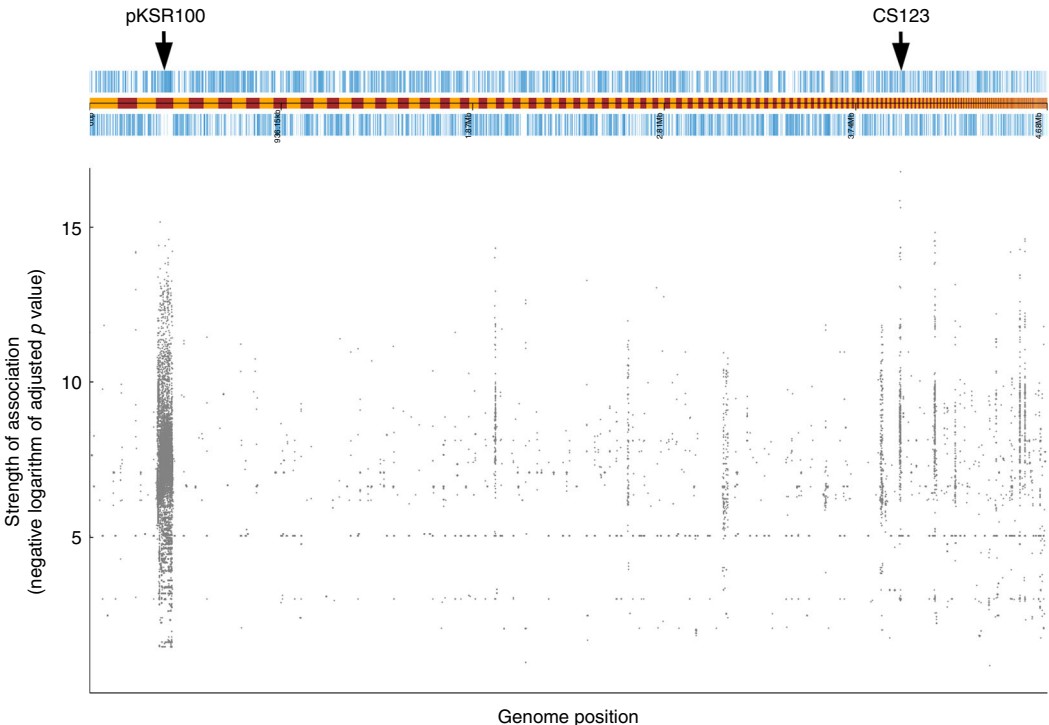

**Fig. 3** Genome-wide association analysis of MSMA sublineages. The draft genome of a *S. flexneri* 2a isolate (ERR1364107) is shown above, with the coding frames on forward and reverse strands of the genome appearing upper and lower to the intervening track (yellow and maroon banded), which shows the boundary of contiguous sequences. The pKSR100 contiguous sequence as well as contiguous sequence 123 (which contains sequence elements of marginally higher significance than pKSR100) are labelled. The Manhattan plot under the genome assembly shows the mapping position of sequence elements with a significant positive-association to MSMA sublineage designation, with the vertical scale representing the negative logarithm of the adjusted *p* value (i.e., the significance) of the association

The exact reason behind the differential population-level success in pKSR100-containing sublineages is unknown. However, the two most likely explanations both have important consequences for public health. First, it is possible that the increased success of pKSR100-containing sublineages is attributable to the increased azithromycin resistance conferred. In which case, these results point to an epidemiologically relevant division of azithromycin resistance for *Shigella* circulating in MSM between the MICs of 32 and 64 mg/l. Alternatively (and not exclusively), is the possibility that the increased success of pKSR100-containing sublineages reflect that different genetic backgrounds of important resistance genes also have a role in determining the population-level success of a pathogen. Differential fitness of mobile genetic elements in bacterial hosts is well established in vitro[27,28], but efforts to link in vitro fitness with epidemiological, population-level outcomes at a hospital-level have been unsuccessful[29]. Although recapitulation of population-level success in vitro is unrealistic, these results frame an exciting line of future in vitrowork with the collection of clinically relevant organisms in our study (and others relevant to HGT of AMR as they arise), that may be useful in interpreting the apparent impact of mobile genetic element carriage in determining pathogen population-level outcomes.

**HGT of pKSR100 among MSMA sublineages**. The detection of the same conjugative plasmid in multi-phyletic sublineages co-circulating a single epidemiological community is strongly suggestive that horizontal plasmid transmission is occurring among *Shigella* MSMA sublineages. The exact route of transfer is unknown, but three non-mutually exclusive scenarios exist: 1 the HGT occurred directly between *Shigella* sublineages in patients simultaneously infected with multiple types; 2 during *Shigella* infection, pKSR100 entered another bacterial host (e.g., in the host microbiota) permitting subsequent transfer to an alternate *Shigella* sublineage on repeat infection of that individual; 3 an alternate bacterial host that contains pKSR100 (possibly acquired during a prior *Shigella* infection) was transmitted onward to another human host who was infected (or subsequently infected) with an alternate *Shigella* sublineage (which then received pKSR100 from the alternate bacterial host). In the first two scenarios, infections of patients with multiple types of *Shigella* is necessary. Among our samples, we found ten patients with repeat *Shigella* infections, including nine with repeat infections of MSMA sublineages, and two patients infected with different species of *Shigella* (suggesting that reinfection with distinct genetic subtypes is occurring, as well as, potentially, chronic infection with near-identical subtypes—see Supplementary Table 4). The third scenario is consistent with the finding of the same (non-pKSR100) azithromycin-resistant plasmid in MSMA *S. sonnei* sublineage 3 and an isolate of verocytotoxin-producing *Escherichia coli* O117:H7 from an outbreak that occurred in the English MSM population in 2014[30] (Methods, Supplementary Fig. 5). Collectively, these findings provide evidence for the plausible epidemiological connectivity and potential mechanisms required for HGT of pKSR100 to have occurred among *Shigella* types in this setting.

In addition to pKSR100 being found in multi-phyletic organisms that were epidemiologically connected, further genomic evidence of HGT of pKSR100 was determined through comparative phylogenetics. Comparing isolate pKSR100 and genome phylogenetic positions indicated both horizontal and vertical modes of plasmid transmission among *Shigella* (Fig. 4). The relative contribution and combination of these two patterns of inheritance varied among the three *Shigella* types. Specifically,

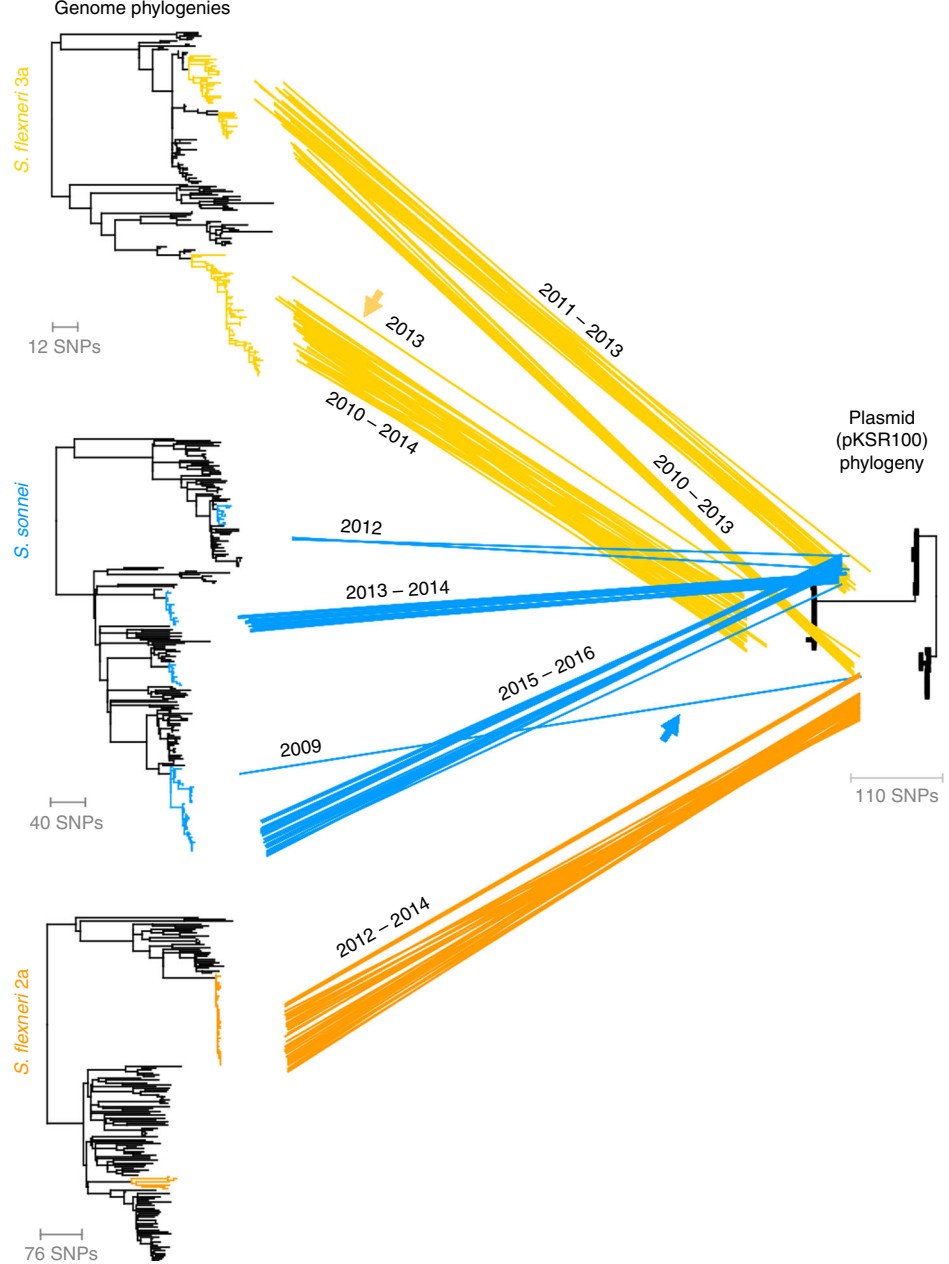

**Fig. 4** Horizontal transmission of pKSR100. Three whole-genome phylogenies for MSMA *S. flexneri* 3a, and *S. flexneri* 2a and *S. sonnei* are shown leftmost with successful and MSMA sublineages (respectively) highlighted in colour. The plasmid tree is a phylogeny of pKSR100 constructed from isolates containing pKSR100. Scale bars in single nucleotide polymorphisms (SNPs) are shown below each phylogeny. Individual isolate genome positions are joined to their plasmid phylogenetic position by intervening coloured lines. Two examples of horizontal transmission are indicated by arrows coloured according to the species. A time-series animation of this figure showing change over time and a static explanatory model are available elsewhere (see: Supplementary Movie 1 and Supplementary Fig. 7)

three successful sublineages of MSMA *S. flexneri* 3a (described in ref. [19]) were associated with three pKSR100 variants (Fig. 4), one of which had an enhanced AMR repertoire, of trimethoprim, sulphonamide, and streptomycin antimicrobial resistances in addition to the ampicillin and azithromycin resistances encoded on all variants (Supplementary Fig. 6). Each plasmid variant was associated with clonal expansions of closely related genomic isolates of *S. flexneri* 3a, consistent with horizontal introduction of the plasmid into each sublineage and successful onward transmission of the sublineage/plasmid combination in the human host (i.e., vertical propagation of pKSR100, Fig. 4). In addition, one *S. flexneri* 3a sublineage isolate had a pKSR100 variant, which was more closely related with pKSR100 found in *S. flexneri* 3a isolates from a different sublineage, which is strongly indicative of horizontal transmission (arrow, Fig. 4). By contrast, *S. sonnei* primarily had one pKSR100 variant across sublineages 1, 2 and 4, and one evolutionary relationship indicated horizontal transmission (arrow, Fig. 4). For *S. flexneri* 2a, there appeared to have been a single introduction of pKSR100 into the major sublineage (Fig. 4).

By examining time alongside the comparative phylogenetics we developed a hypothetical model of the number of HGT events of pKSR100 (Methods, Supplementary Fig. 7), which also enabled correlation of HGT to the enhancement and triggering of

epidemic increases of each *Shigella* type. Although it is possible unobserved organisms are involved in pKSR100 HGT (see above), and our subsample is only a representative quantitation of gene flow, our proposed combined genomic epidemiological model indicated that at least eight horizontal transmission events of pKSR100 occurred among genetically-distinct MSMA sublineages. This hypothetical model is summarised both statically (Supplementary Fig. 7) and as a narrated animation (Supplementary Movie 1), but in brief is as follows: plasmid pKSR100 first appeared in *S. sonnei* in 2009 and from there, moved into an emerging epidemic of *S. flexneri* 3a; pKSR100 then enhanced the epidemic of *S. flexneri* 3a, in which it circulated and diversified into three distinct variants; from there, one pKSR100 variant moved into the major sublineage of *S. flexneri* 2a in 2012, facilitating an epidemic of this sublineage; and another pKSR100 variant moved into (and back into) the three pKSR100-containing *S. sonnei* MSMA sublineages (1, 2 and 4) (Fig. 4).

## Discussion

In capturing the population-level epidemiological consequences of horizontal AMR transmission we have revealed heretofore unseen insights into AMR emergence and spread in important pathogens. Here, horizontal transmission of AMR among pathogenic bacterial species caused fundamental epidemiological shifts, including enhancing emerging epidemics and facilitating epidemics of previously circulating pathogens that were waiting for the 'right' AMR determinant to come along. These dramatic shifts contrasted with the incremental increases in pathogens carrying chromosomally encoded (vertically inherited) AMR, and demonstrated how these two types of resistances (elaborated and adapted to different populations in different geographical areas) can combine to form new highly resistant sublineages. It is important to consider that these overarching observations about AMR emergence were made within a specific patient sub-population known to have increased connectivity and antimicrobial usage (in fact these features were almost certainly prerequisite in our being able to make these observations). However, the genomic processes in play here and their impact on pathogen epidemiology are likely translatable to the broader trends responsible for elaborating the global AMR crisis.

Of course, factors other than AMR, including changes in sexual behaviour and population immunity, will have also shaped these epidemics, and in truth the English MSM shigellosis epidemics are all but gone for now[31]. However, there is a clear and valuable lesson here for future AMR work. It is no longer enough to understand and trace AMR pathogen combinations as simple taxonomic compartments (e.g., azithromycin-resistant *Shigella*), as the genomic diversity contained within such compartments, and genetic exchange among them, also contributes to epidemiological outcomes (particularly where resistance is horizontally acquired). Clearly, we must continue to weave the sub-genomic study of pathogens into all facets of AMR research, surveillance and management, and work to determine the compartments, barriers and quantity of gene flow among pathogenic and non-pathogenic, as well as antimicrobial resistant and antimicrobial susceptible bacterial species.

## Methods

**Demographic associations of MSMA sublineages.** To infer association of sublineages with MSM, the number of patients who were male, who had low-risk travel histories, and who were between 16 and 60 years old were calculated (for each category) and compared with those numbers among the entire (within species) data set among the UK representative subsample. These proportions were then compared using a two-tailed Fisher's exact test (Supplementary Table 2). To generate a single indicative figure of association (i.e., the figure in the results) the same comparison was applied to the proportion of 16–60 years old male patients without a history of recent high-risk travel in MSMA sublineages relative to the

entire UK representative subsample across species (i.e., 159 of 366 total and 96 of 103 among MSMA sublineage isolates).

**Confirmation of sublineage designation as MSMA.** To confirm our classification of phylogenetic sublineages as being MSMA, epidemiological interview data from patients whose isolates were submitted through Institut Pasteur were used to explore the association between recent MSM activity and the phylogenetic classification of isolates. These data are being collected as part of a public health enquiry into an ongoing outbreak of azithromycin-resistant shigellosis in France. Contact details for male patients without a recent history of high-risk travel (i.e., travel to Africa, Asia or Latin America), aged 16–60 years old were retrieved from clinical laboratories and interviewed by telephone by a Santé Publique France epidemiologist (who was blinded to the phylogenetic classification of the patient's isolate). Of 57 patients that fit these criteria, attempts were made to contact 34, and 13 interviews were successfully completed. Interviews comprised a standard questionnaire to retrospectively identify travel history and sexual exposure (sex with male partner(s); sex with female partner(s); sex with both male and females partner(s); no sex partners; unknown; refused to respond) in the month prior to diagnosis. In total, 9 of 10 (90%) patients reporting recent MSM activity within the month prior to diagnosis had isolates belonging to MSMA sublineages and 3 of 3 (100%) patients not reporting such activity had isolates outside of these lineages. This resulted in a significant association of recent patient MSM activity with isolates belonging to MSMA lineages ($p = 0.0140$, two-tailed Fisher's exact test).

**Azithromycin-resistance determinants among MSMA sublineages.** Contiguous sequences were de novo assembled for each isolate using a custom pipeline[32]. Azithromycin-resistance gene-containing contiguous sequences were identified using Resfinder[33]. The length (in bp) of these contiguous sequences was extracted and correlated with phylogenetic position to identify clusters sharing the same azithromycin resistance determinant. Additional AMR genes co-inherited with the azithromycin-resistance genes (Supplementary Data 1) were also used to identify phylogenetic clusters sharing an azithromycin-resistance determinant. The longest length azithromycin-resistance gene-carrying contiguous sequence from each of the MSMA sublineages were compared using BLAST[34] alongside pKSR100 from *S. flexneri* 3a and the related plasmid R100 with visualisation using ACT[35]. Following on from this, Single-Molecule-Real-Time sequencing was performed on 15 isolates, representative from each *Shigella* sublineage as well as through different positions in the pKSR100 phylogeny (results in Supplementary Fig. 6, Supplementary Data 1). DNA was extracted using the MasterPure Complete DNA and RNA extraction kit and sequenced on the Pacific Biosciences Sequel, and the data was assembled as follows: assembly by HGAPv3;[36] circularisation using Circlator v1.1.3;[37] and polishing by Quiver v1 (SMRT-analysis: a software suite for analysing single-molecule, real-time DNA sequencing data, https://github.com/PacificBiosciences/SMRT-Analysis). SMRT-sequenced AZM resistance determinants were unavailable for two sublineage representatives as the *S. sonnei* sublineage 1 representative failed to regrow for extraction, and on comparative genetic analysis, the plasmid from *S. sonnei* sublineage 3 representative had been lost. In the comparative genetic results for these sublineages (Supplementary Fig. 6) the full-length contiguous sequences assembled from Illumina-sequencing are used (and duly indicated).

**Detection of pKSR100 among all isolates.** In addition to the above, a combination of orthologue detection and mapping was used to infer pKSR100 presence across all isolates. Core genome analysis (using roary ref. [38]) of all UK representative subsample isolates and the MSMA *S. flexneri* 3a sublineage was used to identify the presence and count of 86 pKSR100 orthologues among isolates. Those isolates containing ≥60 orthologues were mapped to a concatenated *Shigella* genome and pKSR100 pseudomolecule (as in ref. [19]) and those with ≥90% coverage over the pKSR100 region were deemed to contain pKSR100 (Supplementary Data 1, Supplementary Fig. 3).

**Genome-wide association analysis of MSMA sublineages.** Sequence element enrichment analysis was run on all 357 UK cross-sectional isolates with the 103 isolates belonging to MSMA sublineages being a binary phenotype. Assemblies were generated using a pipeline[32] and kmer counting (of kmers present in between 5 and 95% of isolates) was performed with fsm-lite[39]. Population structure estimation was made using mash[40] and converted into a three-dimensional distance matrix using scripts inbuilt in SEER[39]. Associations of kmers were then run with SEER and filtered for positive effect size; a minor allele frequency of >5%; an unadjusted pvalue of <10e-5; and an adjusted pvalue of 10e-8. Filtered kmers were then converted to fastq format, mapped to a draft assembly of an *S. flexneri* 2a major sublineage isolate, converted to a Manhattan plot (again using SEER inbuilt scripts) and visualised with Phandango[41]. This revealed a significant association of MSMA sublineage designation with pKSR100 (Fig. 3). For comparison of the association of sequence elements detected for pKSR100 and contiguous sequence 123, the positively correlated sequence elements mapping to each of contiguous sequence 4 (pKSR100) and contiguous sequence 123 in the draft genome of isolate ERR1364107 (shown in Fig. 3) were extracted were analysed as follows: the number of sequence elements mapping to each contiguous sequence was divided by the

total length of the contiguous sequence in bases was calculated to generate a 'Positively correlated sequence elements per base' figure; box plots were generated of the negative logarithms of the adjusted p values of the sequence elements mapping to each contiguous sequence; and the means of these data distributions were compared using an unpaired (Welch) t-test (Supplementary Fig. 4).

**Phenotyping of MSMA Shigella.** To determine the phenotype conferred by the various azithromycin-resistance plasmids detected in MSMA sublineages, the MIC of 74 isolates from MSMA and non-MSMA sublineages were determined using Etest strips (BioMérieux) and/or in-agar dilution using a frozen azithromycin reference panel with concentrations ranging from 1 to 256 μg/ml. In-agar antimicrobial susceptibility testing was performed according to Clinical Laboratory Standards Institute methods, with Staphylococcus aureus ATCC 29213/ NCTC 12973 used as a quality control strain[42]. For each isolate, a final inoculum of 5 × 105 CFU/ml was targeted. Plates were read after 18 h of incubation at 35° ± ℃. Results in Supplementary Data 1 and summarised in Supplementary Table 3.

**Plasmid similarity between E. coli and S. sonnei.** An E. coli O117:H7 isolated in 2014 from the outbreak was sequenced using SMRT sequencing as above, and de novo assembly was compared with azithromycin resistance encoding plasmids from MSMA Shigella sublineages in this study. This showed that the S. sonnei sublineage 3 plasmid was present in the E. coli outbreak isolate (Supplementary Fig. 5).

**Comparative phylogenetics of genome phylogenies with pKSR100.** Comparative phylogenetics of pKSR100 with whole-genome phylogenies was conducted within Dendroscope[43]. The whole-genome phylogenies were (from top to bottom): a Bayesian-inferred phylogenetic tree of S. flexneri 3a based on 1243 variable sites (from ref. [19]); the tree presented in Fig. 1 for S. flexneri 2a and the S. sonnei whole-genome phylogeny incorporating the MSMA clusters from the UK in 2015/16 (as shown in Supplementary Fig. 1). The pKSR100 tree was constructed from isolates that contained ≥60 pKSR100 orthologues and mapped to ≥90% of pKSR100 sequence. Mapping and variant calling against pKSR100 (Genbank accession: LN624486) as previously described[19], was followed maximum likelihood phylogenetic inference based on the resulting 110 variable sites using RAxML[44]. The pKSR100 tree in Fig. 4 is rooted to allow visualisation of the HGT events discussed in the text.

**Temporal phylogenetic reconstruction.** Multiple sequence alignments used to generate the comparative phylogenetic trees for S. sonnei, S. flexneri 2a, were analysed by Bayesian Evolutionary Analysis by Sampling Trees (BEAST) as in ref. [19] (three 50,000,000 length chains using the strict molecular clock and exponential population model with a 10% burn-in) and median (Fig. 2) and 95% Highest Posterior Densities (Supplementary Fig. 7) for MSMA and ciprofloxacin-resistant sublineage most recent common ancestors were extracted.

**Ethical statement.** For data relating to isolates from Public Health England: No individual patient consent was required or sought as PHE has authority to handle patient data for public health monitoring and infection control under section 251 of the UK National Health Service Act of 2006 (previously section 60 of the Health and Social Care Act of 2001). The project was reviewed and approved by the PHE Research Support and Governance Office and Caldicott Panel deemed to comply with public health surveillance standards. For data relating to isolates from Institut Pasteur: Data collected were part of an ongoing public health investigation into an ongoing outbreak of azithromycin-resistant shigellosis and as such were not subject to ethical approval (see Methods).

**Data availability.** All sequencing and genome assembly data for isolates in this study are available under project number PRJEB12097 in the European Nucleotide Archive. Further detail linking isolates and sample accession numbers within the project are available in Supplementary Data 1. The authors declare that all other data supporting the findings of this study are available within the article and its Supplementary Information files, or are available from the authors upon request.

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

## Acknowledgements

The authors are grateful to the administrative, sequencing and pathogen informatics teams at the Wellcome Trust Sanger Institute as well as Dr John Lees for discussions regarding genome-wide association study, and other helpful scientists who commented on the work in progress during formal and informal presentations. The authors also thank John Were for his contribution to managing epidemiological data and Vivienne DoNascimento for her technical assistance in susceptibility testing. This work was supported by Wellcome Trust grant number 206194, and KSB is supported by a Wellcome Trust Clinical Research Career Development Fellowship (106690/A/14/Z).

## Author contributions

KSB, TD, NF, GH, CJ, NRT conceived and developed study. KSB and NRT secured funding. KSB directed research, performed analyses and wrote the manuscript drafts. TD, NF, TC, GH and CJ subsampled prepared English isolates for sequencing. FXW and SL selected, susceptibility tested and prepared French isolates for sequencing. NF, HM, GH identified UK patients with repeat infections. MT conducted epidemiological interviews of French patients. MD conducted susceptibility testing of English isolates. All authors contributed to and approved the final manuscript.

## Additional information

**Competing interests:** The authors declare no competing interests.

