## [Peer Review File · Nature Communications]

Reviewers' comments:

Reviewer #1 (Remarks to the Author):

The authors present a thought provoking piece on the potential role of HGT across pathogens that impact MSM populations. I found the piece well written and largely clear, however certain areas would benefit from more detail and clarity.

-More information is needed about where the specimens and data from the UK shigella samples were obtained. Specifically, how was travel history obtained? As written it is not clear how the UK obtained these specimens and the accompanying epidemiologic data. Additionally were all relevant epidemiologic data available on these specimens (i.e., MSM status, antibiotic history, etc). These are critical points necessary for the reader to determine whether there may be any biases in the samples used

-With respect to the shigella specimens that were one offs and those with "re-infections" - were any treatment failures documented? What is the possible role of treatment failures?

-Finally the authors present their analysis in light of "a single epidemiologic transmission network (line 252-3). Yet, this is never supported by any data. Are they assuming that all MSM are infected through one network? Additionally, the paper suggests that because the specimens were from MSM that they were epidemiologically related. However, the STD field, epidemically related would suggest persons naming each other as sexual partners. This does not appear to be the case. The paper must do a better job of tempering its language about what can truly be said epidemiologically about these samples that come from MSM, but not necessarily MSM who have had sexual contact

Reviewer #2 (Remarks to the Author):

This study synergistically applies genomic and classical epidemiology strategies to establish a transmission network of MSMA infections in the UK based on random subsampling of isolates fortified with metadata that allowed the authors to chart the emergence and demise of MSMA Shigella lineages. The authors identified an epidemiological network of concurrently circulating MSMA (*S. flexneri* 2a and *S. sonnei* (1-4)) in the UK, supported by samples and metadata from France. Genomes were further profiled for chromosomal and plasmid-borne resistance loci are linked to therapeutic treatments for shigellosis but also therapeutics against other STIs found in the MSM group. Variable MIC phenotypes for the pKSR100 conferred azithromycin resistance were recorded and correlated to plasmid plasticity. The assessment of population-level fitness as defined in the manuscript as "number of isolates attributable to a given MSMA sublineage per year" is somewhat limited", however the prevalence and distribution of AMR pKSR100 among sublineages support this notion, and other factors potentially shaping the epidemic MSMA spread are discussed.

The authors report a significant correlation of AMR plasmid pKSR100 prevalence and fitness in the epidemiological network as judged by "long term" persistence of the lineages. While the particular evolutionary fitness factor remains unknown, AMR plasmid acquisition seems to coincide with the emergence and long-time persistence of these lineages. Whole genome phylogeny analysis of the core and carried archetypical plasmids genomes allowed retracing the evolutionary history of plasmid acquisition dynamics and identified lateral transfer events. I did not identify any major weaknesses in the study design, analysis or interpretation of data. In outlining the population-level epidemiological impact of AMR acquisition in the MSMA network research findings should appeal to the broader AMR research and public health community.

Please find below my suggestion to improve the manuscript:

Figure 1, 2 and 3 would all benefit from panel (a, b, c) subheading and additional "text" labels for the

ease of navigating the figures, other than stating color codes in the legend, as well as introduction of proper x- and y-axis labels.

Is there any experimental evidence that pKSR100 negative MSMA lineages are susceptible to plasmid acquisition?

Can the authors elaborate on what is meant by "sub-genomic" (line 19, onwards), as the presented results are largely based on sequence-derived information applying genomics epidemiology principles?

Line 227 "it the possibility"

Reviewer #3 (Remarks to the Author):

This manuscript presents very valuable data on outbreaks of antibiotic resistance in *Shigella*. Unfortunately, too much is made about claims of novelty. I do not know if they are true but even if they are it is not that interesting that this is the first. A review would be interesting but that is not the purpose of the manuscript. Secondly it lacks rigour. The authors claim they have measured selection but really have not (this is challenging in bacteria) and I doubt the logic of the GWAS, in any case it needs to be presented what was done and what it means in the main text. There does not seem to be clear quantification of jumps etc. Overall, the manuscript lacks sound quantitative analysis and while some quantification would definitely add value (for example it is claimed that there is a difference between vertical and plasmid inheritance in the abstract, could numbers be put to this?) it in any case belongs in a more epidemiological journal, where it can be presented in a more sensible way.

Author's response: Bold

General:

We would like to thank the reviewers for their time and helpful comments on the manuscript, which we feel has led to its improvement. NB: Line numbers refer to Line numbers in the revised manuscript, except where specified as 'old L', where they refer to the original manuscript submission.

Reviewers' comments: Reviewer #1 (Remarks to the Author):

The authors present a thought provoking piece on the potential role of HGT across pathogens that impact MSM populations. I found the piece well written and largely clear, however certain areas would benefit from more detail and clarity.

-More information is needed about where the specimens and data from the UK shigella samples were obtained. Specifically, how was travel history obtained? As written it is not clear how the UK obtained these specimens and the accompanying epidemiologic data. Additionally, were all relevant epidemiologic data available on these specimens (i.e., MSM status, antibiotic history, etc). These are critical points necessary for the reader to determine whether there may be any biases in the samples used

We thank the reviewer for pointing out this important point, and have provided more explicit information and references on the composition of the archive from which isolates were selected, as well as details of how the epidemiological data was obtained at Ls 125 – 140. The main addition is the following text: “*The national archive contains all S. flexneri and approximately 70% of S. sonnei isolated in hospital, private, and regional laboratories (as onward reference laboratory submission is recommended for S. sonnei, but mandatory for S. flexneri), and basic epidemiological data extracted from sample submission forms i.e. region, gender, patient age, and travel history where noted (as in^{21,22}).*” Further epidemiological and clinical data (such as antimicrobial usage history and sexual practices) was not routinely available, which is the association of MSMA sublineages with MSM activity needed to be confirmed by actively seeking extended data for a subset of samples (see old Ls 120 -123, now Ls 150 – 153 and Methods Ls 413 - 429).

-With respect to the shigella specimens that were one offs and those with "re-infections" - were any treatment failures documented? What is the possible role of treatment failures?

As there was no access to clinical information, the possibility of treatment failures could not be investigated. The reviewer raises an important point that it is possible that some patient re-isolations arose from chronic infections (the serial-sampling of which might have been associated with treatment failures), but this would not have been the case where the serial isolations were distant genetic-relatives (e.g. different species). This distinction was highlighted in the original manuscript (old L693 – 695, now Supplementary Table 5, Ls 858 - 864), but to better facilitate this understanding in the revised manuscript, we have brought this distinction into the main text (see L306 - 308) and added explicit genetic distance information to Supplementary Table 5.

-Finally the authors present their analysis in light of "a single epidemiologic transmission network (line 252-3). Yet, this is never supported by any data. Are they assuming that all MSM are infected through one network? Additionally, the paper suggests that because the specimens were from MSM that they were epidemiologically related. However, the STD field, epidemically related would suggest persons naming each other as sexual partners. This does not appear to be the case. The paper must do a better job of tempering its language about what can truly be said epidemiologically about these samples that come from MSM, but not necessarily MSM who have had sexual contact

The reviewer is absolutely correct, and it was not our intention to imply that there was any (known) direct contact among patients in the study. We merely meant to highlight that these sublineages are co-circulating in time, space, and in an epidemiological community sharing a common route of transmission (i.e. sexual) in geographically concentrated, dense sexual networks characterised by MSM with similar risk profiles (in terms of STI/HIV coinfection, partner numbers, use of smart phone applications and recreational drugs for sexual encounters, etc) which have previously been associated with the other STI outbreaks, in order to demonstrate the plausibility of horizontal transmission of AMR among the pathogens under

study (for which direct contact is not necessary, as discussed at L295 – 311). To clarify this point, we have updated the references and text of the manuscript accordingly, including reconciling the now-appropriate terminology throughout the manuscript (at old Ls 72, 79, 93, 119, 150, 251, new Ls 94, 101 – 102, 106, 111, 116, 158, 296).

Reviewer #2 (Remarks to the Author):

This study synergistically applies genomic and classical epidemiology strategies to establish a transmission network of MSMA infections in the UK based on random subsampling of isolates fortified with metadata that allowed the authors to chart the emergence and demise of MSMA Shigella lineages. The authors identified an epidemiological network of concurrently circulating MSMA (*S. flexneri* 2a and *S. sonnei* (1-4)) in the UK, supported by samples and metadata from France. Genomes were further profiled for chromosomal and plasmid-borne resistance loci are linked to therapeutic treatments for shigellosis but also therapeutics against other STIs found in the MSM group. Variable MIC phenotypes for the pKSR100 conferred azithromycin resistance were recorded and correlated to plasmid plasticity.

The assessment of population-level fitness as defined in the manuscript as “number of isolates attributable to a given MSMA sublineage per year” is somewhat limited, however the prevalence and distribution of AMR pKSR100 among sublineages support this notion, and other factors potentially shaping the epidemic MSMA spread are discussed. The authors report a significant correlation of AMR plasmid pKSR100 prevalence and fitness in the epidemiological network as judged by “long term” persistence of the lineages. While the particular evolutionary fitness factor remains unknown, AMR plasmid acquisition seems to coincide with the emergence and long-time persistence of these lineages. Whole genome phylogeny analysis of the core and carried archetypical plasmids genomes allowed retracing the evolutionary history of plasmid acquisition dynamics and identified lateral transfer events. I did not identify any major weaknesses in the study design, analysis or interpretation of data. In outlining the population-level epidemiological impact of AMR acquisition in the MSMA network research findings should appeal to the broader AMR research and public health community.

Please find below my suggestion to improve the manuscript:

Figure 1, 2 and 3 would all benefit from panel (a, b, c) subheading and additional “text” labels for the ease of navigating the figures, other than stating color codes in the legend, as well as introduction of proper x- and y-axis labels.

Having reviewed the figures carefully, we agree with the reviewer that navigability could be improved, and have made changes to the figures and legends accordingly, including additional labels to Figure 1, restructuring and relabelling Figure 2, (see Figs 1, 2 and 3 and legends). Note: we have not subdivided the figures into A, B and C as we feel that although the figures are complex, they are not composite, so are unsuitable for strict delineation.

Is there any experimental evidence that pKSR100 negative MSMA lineages are susceptible to plasmid acquisition?

Although no experimental work has been done on these organisms to date (other than the initial diagnostics) there is no *in silico* evidence (i.e. through Plasmid Incompatibility group typing) to suggest that this is not possible. These organisms will be being worked with *in vitro* in the future (as alluded to at old L231, new L289).

Can the authors elaborate on what is meant by “sub-genomic” (line 19, onwards), as the presented results are largely based on sequence-derived information applying genomics epidemiology principles?

The use of the term sub-genomic (old Ls 19 and 340, new Ls 39 and 389) refers to genetic features of pathogens beyond their phylogenetic position (i.e. in this case the presence and character of azithromycin resistance plasmids).

Line 227 “it the possibility”

We thank the reviewer for pointing out this typographical error, which has been corrected (L284).

Reviewer #3 (Remarks to the Author):

This manuscript presents very valuable data on outbreaks of antibiotic resistance in *Shigella*. Unfortunately, too much is made about claims of novelty. I do not know if they are true but even if they are it is not that interesting that this is the first. A review would be interesting but that is not the purpose of the manuscript.

While we feel that this novel work is a case of primacy (and that such examples are important for scientific and allied-health communities to justify changes in surveillance policy and scientific research funding), we agree with the reviewer that scientific studies are not interesting based on primacy alone. As such, in response to the reviewer's concern, we have removed claims of primacy from the text as this in no way detracts from the novelty and significance of the work presented. Specifically, two occurrences of the phrase 'for the first time' (old Ls 17 and 319, L39 and L368) have been removed from the manuscript.

Secondly it lacks rigour. The authors claim they have measured selection but really have not (this is challenging in bacteria) ...

The paper is focused on reporting important observations about the predominance and movement of antimicrobial resistance among overlapping pathogen epidemics, including how specific strains have become predominant in the MSM community over time. The strength of the association found necessitates discussion of the use of antimicrobials as a selection pressures linked to strain predominance. However, while this discussion may lead to the inference of selection pressures, at no point do we claim to have measured selection, and have merely presented our results alongside a qualified discussion of a working hypothesis in the context of current literature (old Ls 184 – L190, new Ls 231 - 237). A final sentence in the concluding paragraph of the manuscript, that previously mentioned selection, has also been altered to avoid any misunderstanding (see L388).

... and I doubt the logic of the GWAS, in any case it needs to be presented what was done and what it means in the main text...

The GWAS was performed to provide an additional layer of evidence (in addition to the association already presented in the main text and figures) that the plasmid was associated with MSMA sublineages. The difference with GWAS, compared with solely numerical association, is that it reduces the obfuscation of the association caused by vertical inheritance of the plasmid (as genomic population structure is accounted for in the analysis) and also explores whether other genetic elements (e.g. virulence determinants, other unknown features) are associated with MSMA sublineages. As this was an additional layer of evidence and is complex to explain succinctly, we had originally placed the detail of the GWAS in the Extended Data. However, to address the reviewer's concern, we have now explicitly presented the rationale and findings of the GWAS in the main text (see Ls 246 - 250).

There does not seem to be clear quantification of jumps etc.

An explicit quantification of jumps (horizontal transmissions) was made (see old L289 – 291). However, to make this clearer we have altered the wording at this position (Ls 343 - 346).

Overall, the manuscript lacks sound quantitative analysis and while some quantification would definitely add value (for example it is claimed that there is a difference between vertical and plasmid inheritance in the abstract, could numbers be put to this?)

We have carefully reviewed our manuscript, and feel that the quantitative analysis in the manuscript is accurate and reproducible (see Reviewer 2's comment "I did not identify any major weaknesses in the study design, analysis or interpretation of data"). Regarding the reviewer's specific example of the quantitative difference between vertical and plasmid inheritance from the abstract (old Ls 15 – 17, new Ls 36 – 39); we feel that this was sufficiently covered in the manuscript as it was explicitly presented in the text of the results (old Ls 176 – 177, new Ls 214 - 216), in Figure 2, and available to readers in Table S1.

... it in any case belongs in a more epidemiological journal, where it can be presented in a more sensible way.

We respectfully disagree with the reviewer that the article belongs in a journal of narrower scope and is not presented sensibly. Reviewer 1 calls the manuscript a “thought-provoking piece” and specifically comments on the presentation (“I find the piece well-written”), while Reviewer 2 comments on the broad scope “appeal to the broader AMR research and public health community” and Reviewer 3 notes that it “presents very valuable data”. As noted by Reviewer 2, the work is from an emerging field that “synergistically applies genomic and classic epidemiology strategies”, for which a natural home for publication is still being defined. In any case, Nature Communications “publishes high quality research from all areas of the natural sciences”, so we feel that scope is not a relevant criticism here and comes under editorial remit.

Reviewers' comments:

Reviewer #1 (Remarks to the Author):

All of my concerns have been adequately addressed by the authors

Reviewer #2 (Remarks to the Author):

The authors have addressed all my concerns and incorporated additional information and clarifications that have improved the manuscript.

Reviewer #3 (Remarks to the Author):

I am happy to bow to the opinions of the other reviewers and to the journal editors in terms of appropriateness for Nature Communications, however I still have concerns about scientific validity that need to be addressed prior to publication.

Firstly, the section about population level fitness of MSMA lineages needs to be dropped from the manuscript. The fact that a lineage is common does not mean that it is fit, since it may simply be more common in the gene pool from which these strains came. Chinese people are not fitter than English people simply because there are more of them. In any case, the reasoning is not clear to me.

Secondly, I find the negative language about what other studies have done in the introduction to be just unhelpful and not terribly convincing. It is clearer and should take less words to state what the current study offers in a positive way. I still think that the manuscript is overcomplicated and jargon filled in a way that serves to obscure the points being made.

Third, (this is a new point) the evidence that the plasmid actually transmitted directly between the lineages of Shigella seems to be greatly overstated. The epidemiological context and evidence of mixed infection is very interesting and I agree it is worth highlighting in a positive way but what is very noticeable from the plasmid tree is how much variation there is within the plasmid sequences, which fall into three distinct clades. Unless an explanation can be given for how all of this variation can have accumulated within the plasmid within such a short time, I would say the most likely scenario for the multiple evidence is that selection pressure has been the same on the different MSMA lineages - a new niche of individuals who are receiving similar antibiotic treatment and they have all reacted by importing the same plasmid, but most likely from different sources, not from each other.

Forth I have now looked at the GWAS, which does indeed confirm a strong association of MSMA and the plasmid, which I could probably agree was driven by selection, although I find the current reasoning convoluted and unconvincing. If a new niche arises during the course of a study that favors shigella with the plasmid, then the lineages that acquire the plasmid and invade the niche will look fit. There is no reason to suppose that these lineages are intrinsically fitter. They just got lucky and acquired the plasmid first.

Fifth, the plasmid is not even the strongest GWAS hit. Please discuss the elements with still stronger associations!!! I think the GWAS should be displayed and discussed in the main text, since it is one of the only pieces of actual quantitative analysis in the manuscript.

Sixth, there are no scale bars on any of the phylogenetic plots. This is poor science and in the context of the third point above an important omission.

Reviewer #3

I am happy to bow to the opinions of the other reviewers and to the journal editors in terms of appropriateness for Nature Communications, however I still have concerns about scientific validity that need to be addressed prior to publication.

1. Firstly, the section about population level fitness of MSMA lineages needs to be dropped from the manuscript. The fact that a lineage is common does not mean that it is fit, since it may simply be more common in the gene pool from which these strains came. Chinese people are not fitter than English people simply because there are more of them. In any case, the reasoning is not clear to me.

We disagree with the reviewer on this point and will note at the outset (in case there has been a misunderstanding), that this section discusses the relative fitness of UK MSMA sublineages with each other, not with the remainder of the *Shigella*.

1.1 We have extensively sampled the gene pool from which these strains came, so we have the relevant context in the UK showing the relative contribution of different lineages to that population. It is against this that we are measuring the change in sublineage frequency over time (e.g. shown in Figure 2) which provides evidence for fitness. In contrast to many previous genomic epidemiology studies, our sampling was not-selective and was representative across the entire diversity of *Shigella* in the United Kingdom over the time-period encompassing the epidemics (i.e. including other e.g. non-MSMA *Shigella* and travel related isolates). This ensured that the gene pool from which these strains came was adequately represented among our dataset. Furthermore, the measure of population fitness outlined in the manuscript is not reliant solely on numerical expansion (outlined at Ls 129 – 137). This strong evidence is also complemented with evidence of persistence in time beyond the course of our rigorous sampling window (i.e. MSMA sublineage observation in MSMA outbreaks in the UK in 2015-16, see Ls 137 – 142), and evidence of invasion of a connected, but separate niche (i.e. presence in MSM in France, see Ls 144 – 155).

1.2 The analogy of Chinese and British people is erroneous as the section discusses relative fitness of the MSMA sublineages with each other. Chinese people and English people are not competing in the same way sublineages co-circulating in a single epidemiological community would be.

2. Secondly, I find the negative language about what other studies have done in the introduction to be just unhelpful and not terribly convincing. It is clearer and should take less words to state what the current study offers in a positive way. I still think that the manuscript is overcomplicated and jargon filled in a way that serves to obscure the points being made.

We have edited the text in response to the reviewer's concern. See change at L60 – 64

From:

“Although observations have been made that are suggestive of HGT of AMR (i.e. similar or identical AMR plasmids in multiple bacterial species within a hospital or patient),^{4,5} these are often presented without epidemiological data that plausibly support how HGT

might have occurred, or information on the epidemiological fate (e.g. seeding an outbreak, or a dead-end transmission) of the newly formed pathogen-AMR combination.”

To:

“Observations have been made that are suggestive of HGT of AMR (i.e. similar or identical AMR plasmids in multiple bacterial species within a hospital or patient),^{4,5} and such evidence would be enhanced by strong epidemiological data to support the plausibility of HGT occurring, and information on the epidemiological fate (e.g. seeding an outbreak, or a dead-end transmission) of the newly formed pathogen-AMR combination.”

3. Third, (this is a new point) the evidence that the plasmid actually transmitted directly between the lineages of *Shigella* seems to be greatly overstated. The epidemiological context and evidence of mixed infection is very interesting and I agree it is worth highlighting in a positive way but what is very noticeable from the plasmid tree is how much variation there is within the plasmid sequences, which fall into three distinct clades. Unless an explanation can be given for how all of this variation can have accumulated within the plasmid within such a short time, I would say the most likely scenario for the multiple evidence is that selection pressure has been the same on the different MSMA lineages - a new niche of individuals who are receiving similar antibiotic treatment and they have all reacted by importing the same plasmid, but most likely from different sources, not from each other.

3.1 We heartily agree with the Reviewer that the plasmid may be being transmitted indirectly, as was elaborated in the manuscript where multiple explanations for the appearance of the plasmid in multiple sublineages are considered and discussed. To quote from the original text at Ls 237 – 245:

*“The exact route of transfer is unknown, but three non-mutually-exclusive scenarios exist: (1) the HGT occurred directly between *Shigella* sublineages in patients simultaneously infected with multiple types; (2) during *Shigella* infection, pKSR100 entered another bacterial host (e.g. in the host microbiota) permitting subsequent transfer to an alternate *Shigella* sublineage on repeat infection of that individual; or (3) an alternate bacterial host that received pKSR100 during *Shigella* infection was transmitted onward to another human host who was infected (or subsequently infected) with an alternate *Shigella* sublineage (which then received pKSR100 from the alternate bacterial host).”*

In line with your comments we then outline the evidence to support the possibility of the indirect mechanisms occurring in this setting (i.e. repeat patient infections, and evidence of plasmid sharing with *E. coli*).

3.2 Regarding the amount of variation in the plasmid, it is minimal: 110 single nucleotide polymorphisms in total was seen across the plasmids among 166 isolates. To underline this and to correct an omission (see response to point 6 below) we have inserted scale bars to make this explicit.

4. Forth I have now looked at the GWAS, which does indeed confirm a strong association of MSMA and the plasmid, which I could probably agree was driven by selection, although I find the current reasoning convoluted and unconvincing. If a new niche arises during the course of a study that favors *shigella* with the plasmid, then the lineages that acquire the plasmid and invade the niche will look fit. There is no reason to suppose that these lineages

are intrinsically fitter. They just got lucky and acquired the plasmid first.

We do not argue that intrinsically-fitter lineages entered the new niche of MSM transmission over the course of the study (see response to point 1 where we reiterate that the relative fitness is being measured among MSMA sublineages).

In essence our data shows that (with section subheadings underlined for reference to the manuscript):

- 1. Background: A niche of dense MSM transmission with high antimicrobial use exists**
- 2. Co-circulating MSMA sublineages: Six sublineages of *Shigella* enter this niche for unexplored reasons (unexplored because it was not the aim of this study to investigate why sublineages might perform better in MSM and partly because this process will be partly ‘lucky’, as the reviewer suggests).**
- 3. Population level fitness of MSMA sublineages: Of these six MSMA-sublineages, four appear epidemiologically ‘fitter’**
- 4. Associations with resistance: All six sublineages are associated with azithromycin resistance (evidence supporting 1) but**
- 5. Fit sublineages are associated with pKSR100 only the four ‘fitter’ lineages are associated specifically with plasmid pKSR100.**

So the reviewer’s comment “*There is no reason to suppose that these lineages are intrinsically fitter. They just got lucky and acquired the plasmid first.*” is entirely correct and in the absence of selection this is where it may have ended, but within the MSM community the plasmid is what is associated with lineage success.

Fifth, the plasmid is not even the strongest GWAS hit. Please discuss the elements with still stronger associations!!! I think the GWAS should be displayed and discussed in the main text, since it is one of the only pieces of actual quantitative analysis in the manuscript.

As outlined in the text, the GWAS was conducted primarily to confirm the association of pKSR100 with MSMA sublineages independent of population structure. In doing so, GWAS demonstrated that the plasmid was highly significant, hence our understandable focus on that element. It is true that there was one other contiguous sequence (123) to which three MSMA-associated sequence elements mapped (‘hitting’ an intergenic region, a hypothetical protein and a putative chromosome segregation ATPase) that had slightly higher significance than some of the pKSR100-mapping sequence elements (which was clearly displayed in the Figure, now moved to Figure 4 (with some more labelling and see new legend Ls 634 – 642) of the main text at the reviewer’s suggestion). However, this contiguous sequence had far fewer MSMA-associated sequence elements per unit length, and the average association of the sequence elements was significantly lower than in pKSR100. We agree with the reviewer that this information of interest could be presented in the manuscript without interrupting the narrative. As such, we have described the analysis in the methods (L396 – L404), the result briefly in the text (see L195 – L199), and provided the detailed quantitative and qualitative information regarding comparison of the contiguous sequences contain the most significant sequence elements in the Supplementary information (see Supplementary Figure 4).

Sixth, there are no scale bars on any of the phylogenetic plots. This is poor science and in the context of the third point above an important omission.

We agree this was unclear as the scales were provided as alignment lengths in the figure legends. We have now moved the scale bar into the figures, as suggested.

Reviewers' comments:

Reviewer #3 (Remarks to the Author):

While the authors have made efforts to address several of my less substantial concerns and I do understand that it is natural to be a little dismissive of the "third reviewer" when the other two are happy, there are still major things that I do not like about this study. The underlying problem is that it feels like the authors have started from the premise of writing a "high impact" study and made a bunch of claims based sometimes on quite sketchy reasoning. Then it is the job of reviewers to get them either properly justified or removed one by one, in the face of some stubbornness, especially about the more headline grabbing claims. This is not a healthy approach to science or to publishing and puts too much burden on reviewers. It is not the first time I have felt this about recent manuscripts I have seen from the Sanger Institute. There is a degree of hubris creeping in that leaves a bad taste in the mouth.

The element that bothers me most is the claim that lineages are fit, based only on them being at high frequency. This is not based on anything approaching defensible scientific reasoning and unless this is removed or better justified, the study should be rejected.

I understand that the authors are proud of their sample but good sampling of one country does not constitute good sampling of the gene pool, unless it is shown that transmission between countries is rare, which is the opposite of what I understand from this study. For example, if there was source, sink dynamics going on with US or non-MSMA population as a source and UK as a sink and the major 2a lineage had expanded to high frequency in the US or in the non MSMA source, then this could explain a high frequency of 2a in UK MSMA population, without any selection at all in the UK.

Even without this type of objection however, the reasoning is exceptionally sloppy:

"Taking the number of isolates attributable to a given MSMA sublineage per year as a crude estimate of population-level fitness"

How can a number of isolates be a measure of fitness? 27 isolates/year (for example) is not even in the right units to be a fitness measurement. This is scientifically illiterate.

Does the fitness change every year along with the number of isolates?? etc. etc.

Another objection is that the lineages are branches of the tree that the authors have chosen according to opaque criteria. There are many different ways to split the tree and thus many possible ways to choose lineages. If subsets of the tree with few isolates are chosen as lineages, then they will have low fitness according to this measure. This is bonkers. I am not nearly as clear as I would like to be about how the MSMA-dominated have actually been chosen (for example it would be possible to put raw epidemiological data on the tree to justify this more transparently) but in any case there is no reason at all to treat the lineages as equivalent in a way that allows their fitness to be compared. What actually is the intended logic behind this? I have no idea.

Furthermore, there is a difference between being fit and being lucky. The 2a lineage could have expanded rapidly simply because it infected a single super-spreader (whether a person or a nightclub).

The current reasoning of the authors is like saying that because wherever one goes on holiday, one sees lots of chinese people, chinese people must be fit. chinese people expanded to high numbers in

past centuries due to having large fertile lands and have become able to travel in large numbers all over the world in the last couple of decades due to their increased prosperity but they currently have rather few children. Anyway, the point is the onus is on the authors to make a coherent argument, which they have totally failed to do, not to rebut mine.

If this is published in its present form, from this influential research group, we can expect other authors to make similar claims based on similarly sloppy and opaque reasoning. Allowing this to be published as it stands would set a terrible precedent and example for rigour in bacterial epidemiology, especially regarding claims of fitness.

My second substantial continuing scientific objection is that despite the fanfare (although a bit less than before) about how they have a great context to investigate plasmid transmission, there is still not a plausible analysis, especially about how many transmission events of the plasmid are actually likely to have occurred between the *Shigella* lineages themselves, in the UK or elsewhere.

First, there is no discussion how the diversity of the plasmid amongst *Shigella* in UK relates to the global diversity of the plasmid, both in *Shigella* and (probably just as important) in other hosts. The discussion of transmission in the introduction to the section (highlighted in the rebuttal) puts *Shigella* at center stage in the transmission of the plasmid, but it is just as likely that other more numerous but less pathogenic lineages constitute the bulk of the carriers of the plasmid and that *Shigella* has been infected repeatedly from other sources, without necessarily constituting an important reservoir in itself.

Secondly, now that the tree has a scale bar, it is clear that there is plenty of diversity, with a comparable number SNPs to that found phylogeny of the *Shigella* lineages under investigation (which are several megabase genomes instead of 70kb!!!). Thus 110+ SNPs indicates quite a long history. If we assume equal mutation rates, probably almost a hundred times the history of the *shigella* lineage (which are not estimated). It seems entirely implausible that all of this diversity arose during spread amongst *shigella* during these epidemics, which is what the text and video implies, that all of this diversity arose since 2009. As well as being implausible it seems phylogenetically illiterate. For example, according to the scenario shown in figure S7 lineage i of the plasmid has undergone relatively little evolution (but still quite a lot on a per base pair level!) in the time that lineage ii and iii emerged.

The whole section would be greatly improved if the authors took the trouble to derive a molecular clock, both for the bacteria and the phage and attempt to put the evolution in a proper calibrated context.

The title is silly, except as an expression of Sanger Institute hubris. This is a genetic epidemiology paper about *Shigella*, focussing on a single cohort and it is a disservice to the *Shigella* community, who are the most important consumers of this work but will not be able to find it easily. "Multiple pathogens" seems positively misleading. Multiple lineages of *Shigella*, I could accept.

Re: Revision of manuscript NCOMMS-17-2519B - Horizontal antimicrobial resistance transfer drives epidemics of multiple *Shigella* species

Please find attached our revised version of the above manuscript. We are grateful for your additional consultation with Reviewer #2 and are happy to have made the suggested amendments to the manuscript, which we feel is improved as a result. Your combined feedback offered four concerns to be addressed in a revised version of the manuscript, which have been addressed as below (L#s refer to manuscript version with tracked changes):

1. “Controversial terminology, such as e.g. “population-level fitness” and “bacterial fitness” should be avoided and associated conclusions should be toned down”

To address this concern, we have avoided this term when referring to the population-level indicators (see L131 -179, L724) examined (though retained it when citing literature referring to *in vitro* work, where the term is common and well-defined). Instead, we have used the collective term ‘population-level success’ to describe the three specific epidemiological outcomes measured, and have provided a clear definition of this terminology in the relevant results section (see Ls 131 – 136 “*Although MSMA sublineages were co-circulating their population level outcomes differed. Using the combined genomic and epidemiological information, we defined several markers of population-level success. Specifically, these were: a higher relative disease burden; greater longevity/persistence of a sublineage in the MSM population over time; and evidence of international transmission.*” and amended Figure 2 where *the term ‘other fitness markers’ has been substituted with ‘other population-level outcomes’*). The associated conclusions have also been toned down see changes through (Ls 237 – 261) and deletion at L340.

2. “It may be beneficial to further stress the hypothetical nature of the presented transmission scenario”

We have stressed that the transmission scenario we present is inferred from the data as a possible explanation for the observations captured in the study, including the reiteration of alternative scenarios within the section (see L270 – 272, L322 – 340).

3. “Molecular clock analysis would strengthen conclusions.”

Molecular clock analysis has now been added to the study and indeed strengthens the conclusions of the manuscript. The methods and primary finding (that sublineage most recent common ancestors correlated with the time of first observation of each sublineage) further supports the originally-drawn conclusions and is presented in the manuscript at Ls 123,175 – 176, 512 – 517), and revised versions of Figure 2 and Supplementary Figure 7.

4. “The title should be modified to clarify that *Shigella* sublineages are in the center of this study.”

The title has been altered to make it clear that shigellae are at the heart of the study (see L3).

REVIEWERS' COMMENTS:

Reviewer #2 (Remarks to the Author):

The authors have addressed all my concern and provide additional analyses in support of the proposed evolutionary schema.

REVIEWERS' COMMENTS:

Reviewer #2 (Remarks to the Author):

The authors have addressed all my concern and provide additional analyses in support of the proposed evolutionary schema.

We are delighted that the reviewer is satisfied with the additional analyses and no longer has any concerns regarding the manuscript.